# 🦁 CHIMERA: Revisiting Shortcut Learning by Vision-Language Models for Diagram Understanding

## Abstract

Diagrams convey symbolic information in a visual format rather than a linear stream of words, making them especially challenging for AI models to process. While recent evaluations suggest that vision-language models (VLMs) perform well on diagram-related benchmarks, their reliance on knowledge, reasoning, or modality shortcuts raises concerns about whether they genuinely understand and reason over diagrams. To address this gap, we introduce CHIMERA, a comprehensive benchmark comprising 7,500 high-quality diagrams sourced from Wikipedia; each diagram is annotated with its symbolic content represented by semantic triples along with multi-level questions designed to assess four fundamental aspects of diagram comprehension: *entity recognition*, *relation understanding*, *knowledge grounding*, and *visual reasoning*. We use CHIMERA to measure the presence of three types of shortcuts in visual question answering: (1) the *visual-memorization shortcut*, where VLMs rely on memorized visual patterns; (2) the *knowledge-recall shortcut*, where models leverage memorized factual knowledge instead of interpreting the diagram; and (3) the *Clever-Hans shortcut*, where models exploit superficial language patterns or priors without true comprehension. We evaluate 15 open-source VLMs from 7 model families on CHIMERA and find that their seemingly strong performance largely stems from shortcut behaviors – visual-memorization shortcuts have slight impact, knowledge-recall shortcuts play a moderate role, and Clever-Hans shortcuts contribute significantly. These findings expose critical limitations in current VLMs and underscore the need for more robust evaluation protocols that benchmark genuine comprehension of complex visual inputs (e.g., diagrams) rather than question-answer shortcuts.

## 1 Introduction

Visual language enables communication through structured visual elements such as symbols, icons, and spatial relationships. Diagrams are a fundamental form of visual language, used in domains such as science, education, and engineering to convey complex information compactly and intuitively [Greenspan and Shanker, 2009, Anderson et al., 2011, Zdebik, 2012, Marriott and Meyer, 2012]. Comprehending diagrams requires a wide range of abilities, from basic visual recognition to complex reasoning, making it a particularly challenging task for AI systems [Seo et al., 2014, Kembhavi et al., 2016, Lu et al., 2021]. Understanding how vision-language models (VLMs) interpret and reason over diagrams is thus both conceptually challenging and practically important: it reveals current limitations and guides the design of future multimodal systems [Li, 2023]. While recent VLMs have shown impressive results on diagram-related benchmarks [Xue et al., 2024, Liu et al., 2024b, Bai et al., 2025, Meta, 2024, Google, 2025, Agrawal et al., 2024, Microsoft, 2025], these benchmarks often focus narrowly on performance and lack a structured evaluation of the step-by-step

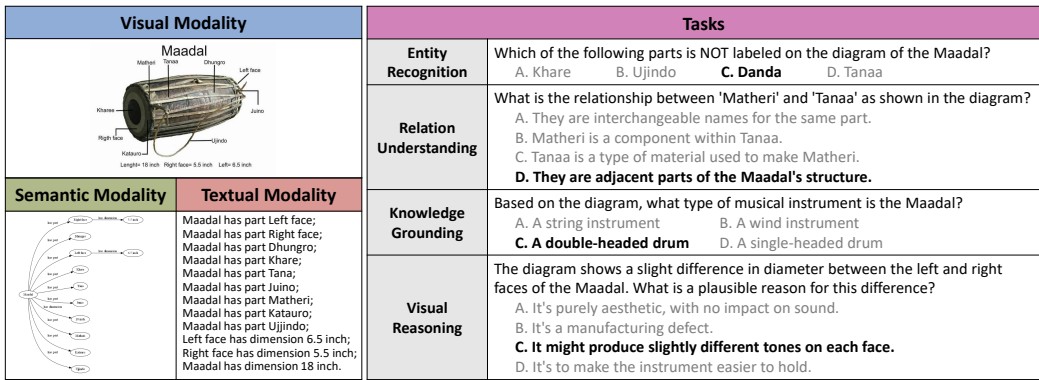

Figure 1: An example from CHIMERA showcasing three modalities (visual, semantic, and textual modality) and four evaluation tasks: entity recognition, relation understanding, knowledge grounding, and visual reasoning.

reasoning process. More importantly, they do not systematically address shortcut behaviors, such as relying on memorized patterns or language priors that can inflate scores without true comprehension [Goyal et al., 2017, Bleeker et al., 2024, Hou et al., 2025]. This highlights the need for a benchmark that not only measures accuracy, but also disentangles how models comprehend diagrams, from basic recognition to abstract reasoning, while controlling for potential shortcuts.

Motivated by semiotics, the study of how meaning is conveyed through signs, we represent the diagram content using semantic triples, enabling consistent alignment across three modalities: the original diagram, i.e., visual modality; visualized triples, i.e., semantic modality; and sentences, i.e., textual modality. Building on Peirce's theory of semiosis, which models interpretation as linking signs to objects through reasoning [Peirce, 1935, Morris, 1938], we frame diagram comprehension as a four-stage process: entity recognition, relation understanding, knowledge grounding, and visual reasoning. This structured perspective reflects the key cognitive steps required for VLMs to move from surface recognition to deeper multimodal understanding.

We introduce CHIMERA, a fine-grained benchmark designed to evaluate the abilities of VLMs to interpret and reason about diagrams with meticulous annotations of both diagram content and evaluation questions. We collect images from Wikipedia [Burns et al., 2023], and clean them using MetaCLIP [Xu et al., 2024a] where unsuitable images such as photos are filtered out. Then, we use VLMs to describe and annotate each diagram with tags of its domain and type, where low-quality images are further filtered out. We further use Gemini [Google, 2024] to describe the essential content that the diagram conveys and use it to annotate semantic triples, and four levels of questions based on the description. We implement consistency checks by running the annotation process multiple times and under different settings to filter out diagrams with low-quality annotations. In total, CHIMERA comprises 7,500 (with 6,000 training set and 1,500 test set) meticulously annotated diagrams, each enriched with a set of semantic triples and four levels of questions targeting entity recognition, relation understanding, knowledge grounding, and visual reasoning (see Fig. 1).

Then, we revisit the shortcut behaviors in visual question answering (VQA) under the diagram comprehension scenario, and categorize them into three distinct types. First, models could rely on image priors, memorizing visual information from training data and using it directly during inference, without genuinely understanding the diagram content [Jayaraman et al., 2024, Li et al., 2024]. We refer to this as the *visual-memorization shortcut*. Second, models could exploit language priors, which we further divide into two subtypes. Given that diagrams often convey factual or domain-specific knowledge, a model could simply recognize high-level visual patterns and rely on pre-trained language knowledge to answer the question without actually understanding the diagram [Hou et al., 2025, Zang et al., 2024]. We refer to this as the *knowledge-recall shortcut*. In addition to that, models can also learn to exploit superficial patterns in the language of the questions or answer options, arriving at correct answers without using the visual input at all [Goyal et al., 2017, Bleeker et al., 2024]. We call this behavior the *Clever-Hans shortcut*, drawing analogy to the phenomenon where models appear to perform well by exploiting spurious cues rather than genuine understanding.

Using CHIMERA, we evaluate 15 open-source VLMs from 7 model families to analyze their core abilities and behavioral patterns in diagram comprehension. We compare model performance on

visual modality and semantic modality. Surprisingly, VLMs perform slightly better on visually complex real diagrams than on the simpler, cleaner semantic graphs. This counterintuitive result suggests that the *visual-memorization shortcut* exists. Models could exploit memorized visual patterns from pretraining, but its impact is **slight**. The *knowledge-recall shortcut* is unlikely to affect entity recognition, but it is more plausible in the remaining three tasks, which are more knowledge-intensive. However, our results show that VLMs perform obviously worse on entity recognition than on the other three tasks, despite it being the simplest and most fundamental. This performance gap supports that the knowledge-recall shortcut occurs **moderately** in the latter tasks. Given that entity recognition is relatively free from knowledge-based shortcuts, we investigate the *Clever-Hans shortcut* in this task. Specifically, we evaluate VLMs without providing the diagram, using only the question and answer options. Surprisingly, some models could even achieve comparable performance as when the diagram is present, suggesting that they rely heavily on spurious linguistic patterns in the prompt. This provides strong evidence that the Clever-Hans shortcut is **significant**.

These findings reveal that the seemingly strong diagram reasoning performance of current VLMs is largely driven by shortcut behaviors rather than genuine comprehension. Among the three types of shortcuts, the Clever-Hans shortcut is the most severe. Our analysis exposes fundamental limitations in current open-source VLMs and underscores the need for more robust evaluation frameworks. Achieving human-level visual understanding remains a long and challenging journey.

## 2 CHIMERA

In this section, we first outline the benchmark design, followed by describing the benchmark construction process in detail and presenting the results of human evaluation.

### 2.1 Design Philosophy: Semiotics and Semiosis

Our data annotation focuses on two key aspects of each diagram: the information content it conveys, and the cognitive abilities required to interpret that information.

**Diagram Information: Three Modalities.** Semiotics, the study of signs and symbols, examines how humans construct and interpret meaning through various forms of representation [Peirce, 1935, Morris, 1938, Cullum-Swan and Manning, 1994]. According to Charles Sanders Peirce and Ferdinand de Saussure, signs are generally categorized into three types: icons, which represent meaning through visual resemblance; symbols, through arbitrary or conventional associations; and indexes, through direct or causal links (e.g., smoke signaling fire) [Peirce, 1935, Yakin and Totu, 2014]. This framework aligns closely with how diagrams convey meaning and how humans interpret them. Inspired by this, we design three modalities in our benchmark that mirror these semiotic types: the visual modality (icon) presents the original diagram; the semantic modality (symbol) visualizes structured triples as graphs; and the textual modality (symbol) expresses them in natural language.[1]

By evaluating how VLMs interpret equivalent content across these modalities, we gain insight into their true comprehension ability. If a model understands the underlying meaning regardless of format as humans do, it should perform consistently across modalities. Thus, grounding our design in semiotic theory provides both a cognitively motivated structure and a principled way to analyze cross-modal reasoning and shortcut behaviors in VLMs.

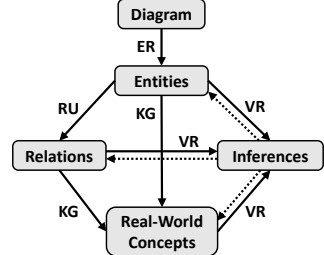

Figure 2: Diagram comprehension process inspired by semiosis.

**Diagram Comprehension: Four Tasks.** Semiosis, as also defined by Charles Sanders Peirce, refers to the dynamic, triadic process through which a sign (e.g., a diagram) represents an object (the realistic entity) and produces an interpretant (the meaning or understanding in the interpreter's mind) [Peirce, 1935, Morris, 1938, Peirce et al., 1992]. This process is iterative, starting with the recognition of signs, followed by the interpretation of the relationships between signs, and then the grounding of their meaning within broader knowledge, which may lead to further reasoning and new insights. Each phase of semiosis is fundamental to fully understanding and reasoning with diagrams.

---

[1]We do not model indexes, as diagrams typically present information explicitly rather than contextually.

Our benchmark tasks are directly aligned with these phases of semiosis (Fig. 2). *Entity Recognition* corresponds to the first step in semiosis, where the model identifies diagram elements (i.e., entities) and associates them with real-world objects. *Relation Understanding* reflects the second phase, requiring the model to interpret the relationships between these entities, understanding how they connect within the diagram. *Knowledge Grounding* involves the model linking its recognized entities and relations to external knowledge, grounding the diagram's information beyond its immediate context. Finally, *Visual Reasoning* mirrors the iterative nature of semiosis, where the model uses its grounded understanding to draw inferences and conclusions, completing multiple cycles of interpretation and reasoning. This framework ensures that our benchmark evaluates the model's ability to process diagrams in a human-like, comprehensive manner.

## 2.2 Benchmark Construction

We build our benchmark in three stages: diagram cleaning, tagging, and annotation (semantic triples and question-answer pairs). An illustration of our construction pipeline is given in Fig. 3.

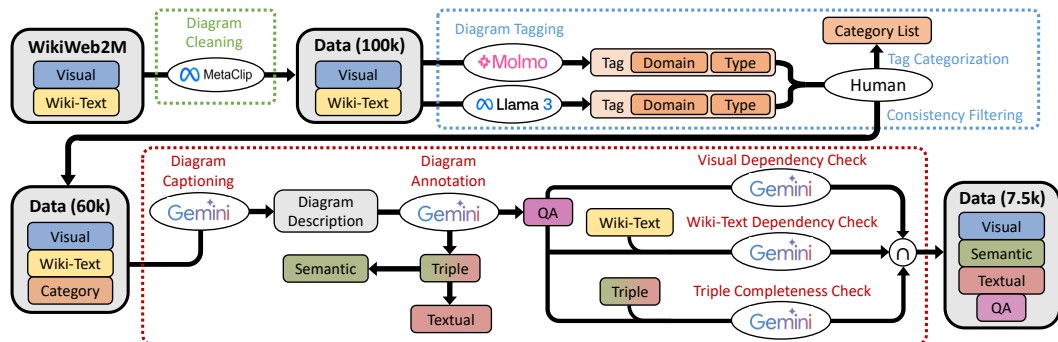

Figure 3: Overview of our benchmark construction pipeline. First, starting from the WikiWeb2M dataset, we use MetaCLIP to remove non-diagram images, resulting in 100k diagrams. Second, we apply Molmo and LLaMA for tagging, and then derive a fixed category list and filter inconsistent results, yielding 60k diagrams. Third, we prompt Gemini to caption diagrams and annotate semantic triples and QA pairs. We then apply three rounds of quality checks, producing a final dataset containing 7.5k high-quality diagrams.

**Diagram Cleaning.** To build our benchmark, we extract images from WikiWeb2M [Burns et al., 2023], a large-scale corpus of English Wikipedia pages. Since many images are irrelevant to diagrams, we apply a filtering process using MetaCLIP [Xu et al., 2024a], combining one positive prompt and six negative prompts. Only images consistently classified as diagrams are retained, resulting in approximately 100k candidate images. Details are provided in App. B.1.

**Diagram Tagging.** Diagrams vary widely in type and domain due to their role in knowledge transfer. To structure our benchmark, we use VLMs (Molmo and LLaMA) to tag each diagram by its type and subject domain (Fig. 3). After aggregating four annotations per image, we group the most common tags into 12 categories across two groups: statistical (e.g., bar chart, line graph) and scientific (e.g., biology, physics). Only diagrams with consistent tags are retained, yielding around 60k images. Full tagging prompts and category details are provided in App. B.2.

**Diagram Annotation.** We posit that the information and knowledge that a diagram conveys can be naturally formalized by a knowledge graph, that is, a set of *semantic triples* [Lassila and Swick, 1999], where each triple contains a head entity, a relation, and a tail entity. In addition to using the diagram as the information carrier (i.e., *visual modality*), we can also represent the information directly by visualizing the semantic triples or transforming it to textual sentences.

Our benchmark includes two core parts of annotations: semantic triples and question–answer (QA) pairs (Fig. 3). To ensure high-quality and consistent annotation, we adopt a two-step pipeline using Gemini-2.0-Flash [Google, 2024] as the annotation backbone. In the first step, we prompt the model to generate a detailed description of each input diagram. These prompts are tailored to different diagram groups and enriched with in-context examples to encourage accurate and specific descriptions. To

reduce hallucinations and improve factual grounding, we also provide the associated Wikipedia text to the model as the supplementary input.

In the second step, we use the generated descriptions to extract semantic triples and generate QA pairs. To ensure that the resulting annotations are both accurate and visually grounded, we apply a three-stage consistency check: (1) we discard examples if questions can be answered without the image; (2) we verify that questions remain unanswerable when only Wikipedia text is available; and (3) we confirm that the semantic triples alone are sufficient to answer the questions. Only diagrams that pass all three checks are retained. After filtering, the final benchmark comprises 6,000 diagrams for training and 1,500 for testing. All evaluations in this paper are conducted on the test set. Additional details, including prompt templates and filtering criteria, are provided in App. B.3.

## 2.3 Human Evaluation

Despite implementing several statistical verification methods to ensure annotation quality, automatically generated annotations may still lack consistency and accuracy. To further assess the reliability of our benchmark, we conduct a round of human evaluation following the automatic annotation process. Unlike the earlier verification, which focused on the independence of Wikipedia text, this evaluation emphasizes the correctness and reliability of the QA annotations. We evaluate each data point along three key dimensions:

- **Visual Dependency**: We assess whether each question truly requires the diagram to be answered, rather than relying on commonsense or background knowledge. An annotation is labeled as *Fully Dependent* if all questions rely on visual content, and *Partially Dependent* if at least one question can be answered without referring to the diagram.

- **QA Correctness**: We evaluate whether the questions are clearly phrased, contextually grounded, and whether the provided answers are correct. Each data point is labeled as *Perfectly Valid* or *Slightly Flawed*, depending on whether any question contains a factual error.

- **Triple Completeness**: We verify whether the annotated semantic triples accurately and sufficiently capture the key information in the diagram. Data points are labeled as *Totally Sufficient* if the triples are complete and correct, and *Marginally Insufficient* if an essential triple is missing or inaccurate.

| Score Ratio (%) | Visual Dependency | | QA Correctness | | Triple Completeness | |
| --- | --- | --- | --- | --- | --- | --- |
| | Fully Dependent | Partially Dependent | Perfectly Valid | Slightly Flawed | Totally Sufficient | Marginally Insufficient |
| Annotator A | 85.3 | 14.7 | 92.0 | 8.0 | 86.0 | 14.0 |
| Annotator B | 100.0 | 0.0 | 99.3 | 0.7 | 80.7 | 19.3 |
| Annotator C | 78.7 | 21.3 | 87.3 | 12.7 | 70.7 | 29.3 |
| Annotator D | 95.3 | 4.7 | 96.0 | 4.0 | 82.7 | 17.3 |

Table 1: Human evaluation results on 300 diagrams across three dimensions: visual dependency, QA correctness, and triple completeness. Scores reflect the percentage of diagrams rated under each category by four annotators (A, B, C, D), showing overall strong annotation quality with minor variations in strictness.

We evenly sample 20% of the test set (300 diagrams) across categories and assign them to four expert annotators (A, B, C, and D). As shown in Tab. 1, the majority of annotations are consistently rated as *Fully Dependent*, *Perfectly Valid*, and *Totally Sufficient*. While minor differences exist among annotators in terms of strictness, the overall results confirm that the benchmark annotations are of high quality and suitable for reliable evaluation.

## 3 Diagram Comprehension Evaluation

In this section, we first present the overall evaluation results on our benchmark. We then delve deeper into a central open question: *how do VLMs actually comprehend complex images such as diagrams?* One hypothesis posits that VLMs achieve genuine understanding, while the alternative suggests that their performance is largely driven by shortcut behaviors. To investigate this, we analyze three typical shortcut types: *visual-memorization shortcut*, *knowledge-recall shortcut*, and *Clever-Hans shortcut* using CHIMERA as a diagnostic tool.

## 3.1 Overall Evaluation

**Experiment Setup.** We evaluate 15 models from 7 model families, covering both academic and industrial models across a range of parameter scales. We select the Qwen2.5-VL (simplified as Qwen) series (3B, 7B, 32B, 72B) [Bai et al., 2025], the LLaMA3.2-Vision-Instruct ((simplified as LLaMA) series (11B, 90B) [Meta, 2024], the Gemma3 series (1B, 12B, 27B) [Google, 2025], the LLaVA-1.6 series (7B, 13B, 34B) [Liu et al., 2024b], as well as three standalone models: Pixtral-12B [Agrawal et al., 2024], Phi-4 5.6B [Microsoft, 2025], and BLIP-3 4B [Xue et al., 2024]. More details about the model, the evaluation setting (e.g., prompts) can be found in App. C.1.

**Overall Results.** We report average accuracy across 15 models in Tab. 2, with detailed results provided in App. C.2. Models are evaluated across three input modalities—visual (original diagram), semantic (visualized triples), and textual (sentence-form triples)—and four tasks: entity recognition (ER), relation understanding (RU), knowledge grounding (KG), and visual reasoning (VR). Overall, VLMs perform best with textual inputs across all tasks, while accuracy drops significantly for visual and semantic modalities, revealing clear room for improvement in diagram comprehension.

| Accuracy (%) | ER | RU | KG | VR |
|---|---|---|---|---|
| Visual Modality | 74.1 | 80.5 | 82.8 | 82.0 |
| Semantic Modality | 70.3 | 79.2 | 83.0 | 80.9 |
| Textual Modality | 88.4 | 90.2 | 91.8 | 88.9 |

Table 2: Average accuracy of 15 VLMs on CHIMERA across three input modalities and four tasks.

## 3.2 Visual-Memorization Shortcut: Do VLMs Answer Using Memorized Visual Patterns?

With the increasing model capacity, recent studies suggest that VLMs could memorize training data (e.g., diagrams) and rely on this memorized content for inference, rather than genuine comprehension [Jayaraman et al., 2024, Li et al., 2024]. We refer to this behavior as the *visual-memorization shortcut*, where a model bypasses reasoning by exploiting memorized visual patterns.

**Experiment Design.** To investigate whether VLMs rely on the visual-memorization shortcut for diagram comprehension, we leverage the multimodal design of CHIMERA. Each diagram in the benchmark is annotated with semantic triples, which are visualized as semantic modality inputs, i.e., structured and simplified versions of the original diagrams. Compared to real diagrams (visual modality), semantic graphs eliminate noise and layout ambiguity, offering a clearer path for reasoning.

If a model is not relying on memorized visual patterns, we would expect it to perform worse on real diagrams than on the cleaner, more structured semantic modality. In contrast, if the visual-memorization shortcut is in use, models might perform better on the visual modality, indicating reliance on memorized diagram appearances rather than actual visual reasoning. Additionally, we treat the textual modality (i.e., sentences generated from triples) as an upper-bound reference, since it presents all essential information in the most language-friendly form for VLMs.

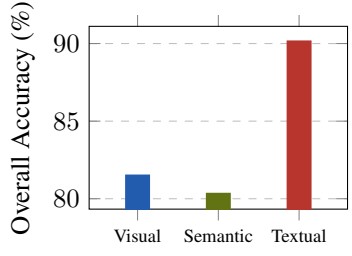

Figure 4: Average performance across models and tasks on different modalities. The overall performance on the visual modality is **slightly** better than that on the semantic modality.

**Evaluation Results.** Fig. 4 reports the average accuracy across all tasks and models. Detailed results are in App. C.2. As expected, performance on the textual modality is the highest, confirming the language-centric nature of current VLMs. However, a surprising pattern emerges: models perform slightly better on the visual modality than on the semantic modality, comparing to the gap between textual modality and the visual modality. Despite being more complex and less structured, real diagrams yield better performance than their simplified semantic counterparts. This contradicts the intuition that structured, noise-free inputs should facilitate better reasoning.

**Takeaways.** These results suggest that VLMs do make **slight** use of the visual-memorization shortcut when performing diagram comprehension. While the relative gap is not large, the fact that models outperform on real diagrams despite their complexity implies some level of visual

memorization. The shortcut effect appears limited but measurable, and it could become more pronounced in settings where training and evaluation data overlap.

### 3.3 Language Shortcuts

In addition to relying on visual memorization, VLMs may also exploit shortcuts derived from the language prior patterns and knowledge embedded in the language modeling component rather than performing genuine multimodal reasoning. We divide such language-based shortcuts into two distinct types: (1) The *knowledge-recall shortcut*, where models retrieve factual or commonsense knowledge from pretraining to answer questions, bypassing the diagram. (2) The *Clever-Hans shortcut*, where models rely on superficial linguistic patterns in questions or answer options, independent of any grounded understanding. In this section, we analyze these two shortcuts in turn.

#### 3.3.1 Knowledge-Recall Shortcut: Do VLMs Answer Using Memorized Knowledge?

A common form of language-based shortcut is the knowledge shortcut, where VLMs draw on memorized background knowledge or commonsense associations from pretraining instead of interpreting the visual content [Hou et al., 2025, Zang et al., 2024].

**Experiment Design.** To assess the presence of knowledge shortcuts, we analyze VLM performance across the four tasks in CHIMERA: entity recognition (ER), relation understanding (RU), knowledge grounding (KG), and visual reasoning (VR). As the most fundamental and prerequisite step in diagram comprehension (Fig. 2), The entity recognition task is highly localized and visual, making it unlikely to benefit from knowledge-recall shortcuts. In contrast, other three tasks involve deeper reasoning and are more likely to draw on factual knowledge stored in the model. Intuitively, if a model engages in genuine visual comprehension, we would expect the highest accuracy on entity recognition, followed by decreasing performance on the more complex tasks. However, if a model performs worse on the recognition but better on other tasks, it suggests a reliance on memorized knowledge rather than true visual understanding, an indicator of knowledge-recall shortcuts.

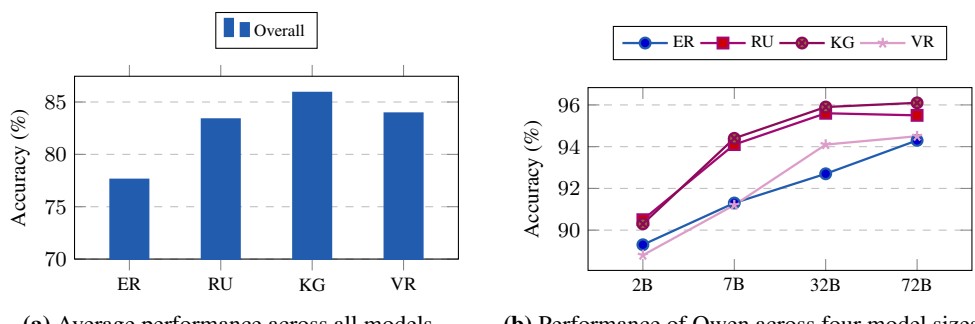

**(a)** Average performance across all models.   **(b)** Performance of Qwen across four model sizes.

Figure 5: The overall evaluation accuracy for 15 VLMs and the accuracy of four Qwen models on the four tasks. VLMs perform on entity recognition much worse than that on the other three tasks. For Qwen models, larger model is more likely to have lager gap between entity recognition and other tasks.

**Quantitative Results.** As shown in Fig. 5a, VLMs surprisingly perform worst on entity recognition, while achieving higher accuracy on relation understanding, knowledge grounding, and visual reasoning. This contradicts the intuition that simpler, recognition-level tasks should be easier. The pattern suggests that VLMs rely on memorized knowledge to handle semantically richer tasks, rather than building understanding through visual parsing. Furthermore, as shown in Fig. 5b, this trend holds consistently across the Qwen model family (from 3B to 72B), with larger models often exhibiting more pronounced gaps. This indicates that larger VLMs are more likely to be susceptible to knowledge-recall shortcuts, likely due to their stronger memorization capacity.

**Qualitative Evidence.** Fig. 6 illustrates a representative failure case from LLaMA-90B. The model incorrectly classifies a scatter plot as a line graph, i.e., failing in basic visual recognition, yet proceeds to correctly describe complex trends in the data and even offer projections and possible data sources.

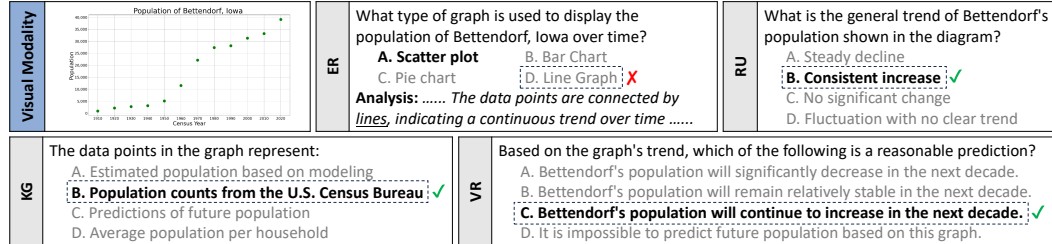

Figure 6: Model responses for a diagram of the largest evaluated VLM (i.e., LLaMA-90B). The model fails to recognize the basic, simple elements in the diagram while providing correct answers for more complex questions.

This behavior reinforces the hypothesis that the model bypasses perception and relies instead on memorized knowledge patterns to perform diagram comprehension.

**Takeaways.** Both quantitative trends and qualitative examples support the conclusion that knowledge-recall shortcuts occur **moderately** in current VLMs. These shortcuts are observed across model sizes and tend to be more pronounced in larger models. While they help models answer knowledge-intensive questions, this often comes at the expense of genuine visual comprehension.

### 3.3.2 Clever-Hans Shortcut: Do VLMs Rely on Superficial Language Patterns?

Another widely observed form of shortcut in visual question answering is the Clever-Hans shortcut, where models exploit superficial patterns in the input text (i.e., the question and answer options), rather than relying on visual input [Goyal et al., 2017, Agrawal et al., 2018, Cadène et al., 2019, Bleeker et al., 2024]. This shortcut is particularly insidious because the model can appear accurate by exploiting linguistic regularities, even when the visual input is missing or irrelevant.

**Experimental Design.** To isolate the Clever-Hans shortcut from other language priors (e.g., factual knowledge), we focus on the entity recognition task in CHIMERA. Our earlier analysis shows that this task is less influenced by the knowledge-recall shortcut, making it an ideal case for probing the effects of shallow language pattern exploitation.

We compare model performance under two conditions: (1) the standard setting with access to the original diagram, and (2) a blank-image setting where no visual information is provided. Since each question in CHIMERA is multiple-choice with four options, the expected accuracy from random guessing is approximately 25%. Any significant improvement above this baseline in the absence of visual input suggests the presence of Clever-Hans behavior.

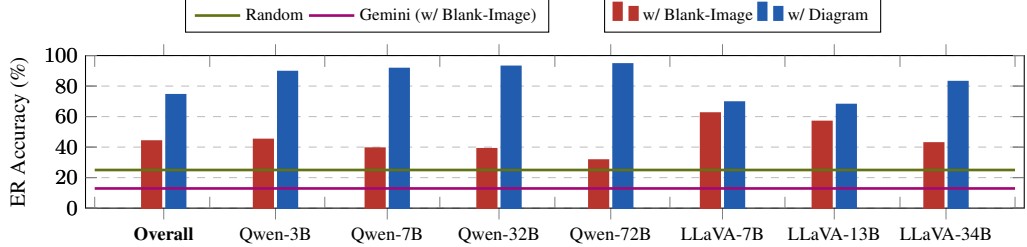

Figure 7: Entity recognition accuracy under normal VQA and blank-image settings. Results show that LLaVA and Qwen models have strong reliance on language-only cues, especially LLaVA models. Besides, larger models exhibiting slightly less susceptibility to the Clever-Hans shortcut.

**Quantitative Results.** Fig. 7 presents entity recognition accuracy across VLMs under both settings. LLaVA models achieve surprisingly high performance even without access to the diagram, for instance, LLaVA-7B reaches over 60% accuracy with a blank image. This strongly indicates that these models rely on language-only cues embedded in the question and options. Qwen models exhibit similar tendencies, although to a lesser extent. Interestingly, we observe that larger models tend to rely less on the Clever-Hans shortcut. For example, Qwen-72B shows a worse performance under the

w/ blank-image setting compared to Qwen-3B. This trend suggests that increased model capacity may improve multimodal grounding, making models more reliant on actual visual content.

**Qualitative Results.** Fig. 8 presents a representative example from LLaMA-90B on the entity recognition task. When the diagram input is removed, the model still generates a confident and contextually reasonable answer by relying solely on the question phrasing and the content of the answer options. Notably, the response lacks any reference to visual content or spatial cues, indicating that the model is not engaging in genuine diagram interpretation. Instead, it is leveraging superficial language patterns, a clear instance of the Clever-Hans shortcut, highlighting its dependence on linguistic biases rather than multimodal understanding.

**Takeaways.** Taken together, these results provide strong evidence that Clever-Hans shortcuts are **significant** in open-source VLMs, particularly among smaller models. Even without valid visual input, models achieve non-trivial accuracy by exploiting linguistic biases . While larger models show some improvement in resisting this behavior, the shortcut remains a significant barrier to robust multimodal reasoning. Addressing it will require improved training signals, more carefully designed datasets, and evaluation protocols that explicitly discourage reliance on language-only cues.

Figure 8: Response of LLaMA-90B on the entity recognition task. Even without a valid diagram input, the model examines the question and options and makes an educated guess based on superficial language patterns.

## 4 Conclusion

We introduce CHIMERA, a comprehensive benchmark for diagram comprehension in VLMs, with structured semantic triples and multi-level tasks. Unlike prior work, it enables fine-grained analysis across modalities and diagram comprehension stages. Our evaluation of 15 VLMs reveals that much of their success stems from language-based shortcuts, especially Clever-Hans behaviors, rather than genuine diagram understanding. These insights highlight key limitations in current open-source models and offer guidance for building more robust, interpretable, and multimodal systems.

## Broader Impact

Structured diagram data holds broad potential for advancing multimodal intelligence across both research and applied domains. The semantic annotations in our benchmark, particularly the structured triples and multilevel reasoning tasks, can support a variety of downstream applications beyond evaluation. For instance, they can enable better text-to-diagram generation, where structured content such as sentences or knowledge graphs can be translated into meaningful visualizations for education, publishing, or user interfaces. Moreover, the design of our benchmark, particularly its explicit separation of reasoning stages and alignment with semiotic principles, can inspire new training paradigms, such as the use of synthetic reasoning trajectories or modality-controlled supervision to improve multimodal model robustness and interpretability. We anticipate that these ideas will generalize to other structured domains, such as scientific visualization, instructional materials, and interactive agents grounded in visual knowledge.

## Limitations

While we offer a comprehensive benchmark for diagram comprehension, several limitations remain. First, our dataset is constructed from Wikipedia diagrams, which, while diverse and high-quality, may not fully represent diagrams used in other domains such as medicine, engineering, or early education. This could limit generalization to domain-specific use cases. Second, although we implement rigorous consistency checks and conduct human evaluation on a subset of the data, automatic annotations, especially for complex reasoning questions, may still contain subtle noise or bias. Finally, while we identify and analyze shortcut behaviors, our diagnostic framework is correlational and does not isolate causal mechanisms behind model behavior. Future work could extend this analysis with counterfactual interventions, synthetic control diagrams, or fine-grained behavioral probing.

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

## A Related Works

**Diagram Question Answering (DQA).** Diagram Question Answering (DQA) is a specialized subfield of Visual Question Answering (VQA), where the input image is a schematic, symbolic, or abstract diagram rather than a natural scene [Hou et al., 2025]. These diagrams commonly convey structured, domain-specific knowledge—such as scientific processes, mathematical relations, or logical systems—making DQA a valuable testbed for evaluating a model's ability to perform symbolic interpretation and structured visual reasoning.

**Benchmarks on Statistical and Analytical Diagrams.** One major category of DQA benchmarks focuses on statistical or analytical charts, such as bar graphs, line plots, and scatter plots. These tasks require models to extract numerical values, recognize trends, and reason over structured visual features. Notable datasets in this area include FigureQA [Kahou et al., 2018], DVQA [Kafle et al., 2018], PlotQA [Methani et al., 2020], ChartQA [Masry et al., 2022], MMC [Liu et al., 2024a], ChartBench [Xu et al., 2024b], and CharXiv [Wang et al., 2024].

**Benchmarks on Visually Structured Content.** Another category evaluates visually structured content, particularly infographics and document-like formats. These include images such as posters, book covers, webpages, and scientific figures, where layout-aware reasoning is critical. Datasets like OCR-VQA [Mishra et al., 2019], DocVQA [Mathew et al., 2021b], InfographicVQA [Mathew et al., 2021a], VisualMRC [Tanaka et al., 2021], and VisualWebBench [Liu et al., 2024c] target the integration of visual structure and textual information.

**Benchmarks from Educational and Instructional Diagrams.** Several DQA benchmarks are derived from science education and domain-specific instructional content, often sourced from textbooks or learning platforms. These diagrams are rich and require external knowledge integration. Key datasets in this space include AI2D [Kembhavi et al., 2016], FoodWebs [Krishnamurthy et al., 2016], TQA [Kembhavi et al., 2017], VLQA [Sampat et al., 2020], and ScienceQA [Lu et al., 2022].

**Benchmarks on Synthetic and Abstract Diagrams.** A final class of benchmarks uses synthetic or abstract diagrams to isolate core reasoning skills. These datasets typically involve geometric primitives or symbolic representations that are free from real-world biases. NLVR [Suhr et al., 2017] and ShapeWorld [Kuhnle and Copestake, 2017] focus on compositional and spatial reasoning, while Zhang et al. [2016] and IconQA [Lu et al., 2021] test high-level relational and symbolic inference through minimalistic, abstract scenes.

## B Details of Benchmark Construction

### B.1 Diagram Cleaning

To construct a comprehensive diagram benchmark, we source images from one of the largest open-source knowledge bases: Wikipedia. Specifically, we use WikiWeb2M [Burns et al., 2023], a large-scale dataset containing over 2 million English Wikipedia webpages with diverse images, rich textual content, and structured metadata.

However, WikiWeb2M includes many non-diagram images such as human portraits, logos, and natural scenes. To isolate true diagrammatic content, we design a binary classification pipeline based on MetaCLIP [Xu et al., 2024a]. We construct one descriptive prompt to identify diagrams and six complementary prompts to exclude non-diagram content. Each image is evaluated across these prompts, and only those classified as diagrams in all negative prompt settings are retained. This conservative strategy ensures high precision in diagram selection. The full list of prompts used in this filtering process is provided in Fig. 9. After filtering, we retain approximately 100,000 diagram candidates for further processing.

### B.2 Diagram Tagging

Since diagrams serve as versatile tools for knowledge transfer, they span a wide variety of types and subject domains. To better organize our benchmark and support structured annotation, we use two vision-language models (Molmo-7B and LLaMA-3.2-7B) to tag each diagram with both its type and

associated knowledge domain (Fig. 9). The full prompt templates used for tagging are available in Figs. 10 to 12.

We repeat the tagging process twice with both models, resulting in four independent annotations per image. We then manually analyze the distribution of tags and consolidate the most frequent ones into 12 categories. These are divided into two groups:

- **Statistical Group**: Includes four types of statistical diagrams — Bar Chart, Line Graph, Pie Chart, and Map.

- **Scientific Group**: Includes eight types of non-statistical diagrams categorized by academic disciplines — Biology, Chemistry, Computer Science, Mathematics, Physics, Astronomy, History, and Music.

To ensure label consistency and reliability, we retain only diagrams with consistent tags across all four annotations. This filtering results in a curated set of approximately 60,000 diagrams.

### B.3 Diagram Annotation

Our benchmark contains two core forms of annotation: semantic triples and question–answer (QA) pairs, which together capture both the content of the diagram and the levels of comprehension required.

To ensure annotation quality, we use Gemini-2.0-Flash [Google, 2024] as the primary annotation model in a structured two-step process.

**Step 1: Diagram Description.** To simplify the downstream annotation and improve quality, we first prompt Gemini to generate a detailed description of each diagram. This intermediate step provides a structured foundation from which semantic triples and QA pairs are derived. Since triple extraction and QA generation emphasize different semantic aspects of a diagram, the description prompts are carefully designed to highlight relevant content.

To reduce hallucination—an inherent issue in large models [Li et al., 2023, Leng et al., 2024]—we supplement each image with its corresponding Wikipedia text to provide factual grounding. Moreover, we design tailored prompts for different diagram groups (e.g., statistical vs. scientific) and include in-context examples to guide the model away from vague or generic outputs. Full prompt details are in Figs. 13 to 16.

**Step 2: Semantic Triples and QA Pairs.** Using the diagram description, we prompt Gemini again to extract semantic triples and generate multiple-choice QA pairs. Detailed prompt designs are available in Figs. 17 to 20.

To ensure the quality of the QA annotations, we implement a three-stage consistency check:

- **Visual Dependency Check (No Image)**: The model attempts to answer questions without seeing the diagram. If it succeeds, the question likely does not depend on the visual content.

- **Wiki-Text Independency Check (No Image + Wiki-Text)**: The model is shown the Wikipedia context but not the image. The question should remain unanswerable.

- **Triple Completeness Check (No Image + Triples)**: The model is given only textual sentences derived from the semantic triples. The question should be answerable in this setting.

Each setting is evaluated twice with shuffled answer choices to minimize bias. We consider a diagram as "succeeded" if the model selects the correct answer in both runs, and as "failed" if it make mistakes in either run.

We discard diagrams:

- That succeed in the entity recognition task in the first two checks, indicating that the QA annotation is not image-dependent.

- That fail in any of the four tasks (ER, RU, KG, VR) in the third check, indicating that triples are incomplete.

After applying these filters, we retain a total of 7,500 diagrams, though the category distribution remains imbalanced. From this pool, we curate a balanced test set of 1,500 diagrams and a training set of 6,000 diagrams. Comprehensive category-wise statistics are presented in Tab. 3. [2]

| Category | Test Set | Training Set |
|---|---|---|
| Bar Chart | 150 | 900 |
| Line Graph | 150 | 350 |
| Pie Chart | 150 | 0 |
| Map | 150 | 2000 |
| Biology | 150 | 900 |
| Chemistry | 150 | 1600 |
| Computer Science | 150 | 0 |
| Mathematics | 150 | 150 |
| Physics | 150 | 100 |
| Others | 150 | 0 |

Table 3: Number of diagrams per category in the test dataset and training dataset.

# C  Supplementary Results

## C.1  Experiment Setup Details

### C.1.1  Model List

We evaluate a diverse set of vision-language Models (VLMs) on our benchmark. Our selection encompasses both industry-developed models from leading AI companies such as Google, Meta, Alibaba, and Microsoft, as well as representative open-source models from the academic community. For certain model families, we include multiple variants with different parameter scales to facilitate comparative analysis. The following models are evaluated in our benchmark.

**Qwen-2.5-VL**    [Bai et al., 2025] is a multimodal model series developed by Alibaba, featuring a native dynamic-resolution Vision Transformer with window attention, enabling efficient processing of high-resolution images and long-form videos. It supports precise object grounding with absolute coordinates and demonstrates strong capabilities in document parsing, chart interpretation, and temporal event localization. In our experiments, we evaluate four variants of Qwen2.5-VL with 3B, 7B, 32B, and 72B parameters.

**LLaMA-3.2**    [Meta, 2024] is a large-scale foundation model family developed by Meta. It introduces multimodal capabilities, integrating image, video, and speech understanding via modular adapters. For vision, it employs a pretrained image encoder, connected to the language model through a cross-attention-based vision adapter. This compositional setup allows the system to process image-text pairs without modifying the core language model. In our experiments, we evaluate four variants of LLaMA-3 with 11B, and 90B parameters.

**Gemma-3**    [Google, 2025] is a multimodal model series developed by Google DeepMind, supporting vision, long-context reasoning, and multilingual understanding. It adopts a decoder-only architecture with grouped-query attention and introduces a local-to-global attention mechanism to reduce KV-cache memory overhead during long-context inference. For vision processing, it can handle flexible image resolutions. In our experiments, we evaluate three variants of Gemma-3 with 1B, 12B, and 27B parameters.

**Pixtral**    [Agrawal et al., 2024] is a multimodal language model developed by Mistral. It features a custom vision encoder trained from scratch, capable of ingesting images at their native resolution and aspect ratio, and supports flexible tokenization strategies. The model employs RoPE-2D position encoding in the vision encoder and uses a decoder-only architecture based on Mistral NeMo. In our experiments, we evaluate the 12B variant.

---

[2] Our data license is CC-BY-4.0.

**Phi-4** [Microsoft, 2025] is a multimodal model developed by Microsoft, extending the Phi-4 series to support text, vision, and speech/audio modalities. It employs a novel Mixture-of-LoRAs architecture that integrates modality-specific adapters without modifying the frozen language backbone, thus preserving its strong language capabilities. In our experiments, we evaluate the 5.6B variant.

**BLIP-3 (xGen-MM)** [Xue et al., 2024] is a multimodal model series developed by Salesforce, designed to unify training objectives and scale vision-language understanding through a simplified architecture. The framework replaces the Q-Former in previous models with a scalable perceiver resampler, enabling efficient any-resolution vision token sampling and supporting interleaved multimodal inputs. In our experiments, we evaluate the 4B variant.

**LLaVA-1.6** [Liu et al., 2024b] is a multimodal model series that enhances visual reasoning, OCR, and world knowledge while maintaining a lightweight architecture. It introduces higher input resolutions and refined visual instruction tuning, enabling better understanding of complex visual scenes. In our experiments, we evaluate three variants of LLaVA-1.6 with 7B, 13B, and 34B parameters.

### C.1.2 Prompt Pipeline

For question answering, we design a three-step, systematic, rule-based evaluation pipeline. In the first step, the model is presented with the input multimodal data and a corresponding question, and is prompted to analyze and answer the question in a step-by-step manner. In the second step, given the full preceding context, the model is instructed to produce a final, conclusive answer in the form of a multiple-choice selection (i.e., A, B, C, or D). To address potential limitations in instruction-following abilities (especially in smaller models), we introduce a third step that automatically extracts the final answer from the model's generated response in Step 2. This is achieved using a set of robust regular expressions and response-processing workflows that identify key phrases, such as numeric values and conclusion markers, to ensure accurate answer extraction and matching. An example of the three-step pipeline is shown in Fig. 21.

### C.1.3 Human Evaluation Guidelines

The guideline for the human evaluation of the data annotation quality assessment is given below.

- **Visual Dependency.** Evaluate whether answering the questions requires visual reference to the diagram. *Fully Dependent* means all questions rely on visual information (e.g., labels, layout, spatial structure). *Partially Dependent* indicates that **at least one question** could be answered without seeing the diagram, using commonsense or background knowledge.

- **QA Correctness.** Assess the overall quality of the four QA pairs. *Perfectly Valid* means all QA pairs are accurate, clear, and grounded in the diagram. *Slightly Flawed* means **at least one QA pair** contains minor issues such as ambiguity, hallucination, or poor phrasing.

- **Triple Completeness.** Examine how well the knowledge triples represent the information in the diagram. *Totally Sufficient* indicates that the triple set is comprehensive, factually correct, and well-structured. *Marginally Insufficient* means **at least one triple** is missing important details, include minor errors, or lack clarity.

### C.1.4 Project Cost

In our benchmark, most experiments are conducted on NVIDIA GPUs, including RTX 3090 and A100, with the specific hardware selected based on model size. For Llama-3.2-90B only, we leverage the Together AI inference API to perform evaluation. Additionally, since we only perform inference on VLMs, we use `torch.bfloat16` precision for all tasks for reducing GPU memory usage.

We report the computation resources to clean and annotate our benchmark. Besides, we report the computing cost for our evaluation. We measure the computation cost by GPU Hours and the financial cost for API models in Tab. 4.

### C.2 Detailed Results

| Task | Model | Data | Type | Cost |
|---|---|---|---|---|
| Diagram Cleaning | MetaCLIP | 2M | H100 | 200 GPU hours |
| Diagram Tagging | Molmo & LLaMA3.2 | 100k | RTX3090 | 400 GPU hours |
| Diagram Annotation | Gemini | 60k | Google API | 8,000 USD |
| Consistency Checking | Gemini | 60k | Google API | 12,000 USD |
| Benchmark Evaluation | 14 VLMs | 1.5k | RTX3090/A100 | 100 GPU hours |
| | LLaMA-90B | | TogetherAI API | 400 USD |

Table 4: The cost of building our benchmark and evaluation on our benchmark.

| Model | Visual Modality | | | | Semantic Modality | | | | Textual Modality | | | |
|---|---|---|---|---|---|---|---|---|---|---|---|---|
| | ER | RU | KG | VR | ER | RU | KG | VR | ER | RU | KG | VR |
| Qwen2.5-3B [Bai et al., 2025] | 89.3 | 90.5 | 90.3 | 88.8 | 88.0 | 90.7 | 93.7 | 89.8 | 89.4 | 92.9 | 91.8 | 88.7 |
| Qwen2.5-7B [Bai et al., 2025] | 91.3 | 94.1 | 94.4 | 91.2 | 87.3 | 93.2 | 95.3 | 90.7 | 90.9 | 93.7 | 95.1 | 91.1 |
| Qwen2.5-32B [Bai et al., 2025] | 92.7 | **95.6** | 95.9 | 94.1 | **93.6** | 95.6 | 97.4 | 95.7 | 95.1 | 96.7 | **98.3** | **96.1** |
| Qwen2.5-72B [Bai et al., 2025] | **94.3** | 95.5 | 96.1 | **94.5** | 91.1 | 94.9 | 97.3 | 95.2 | 95.5 | **97.3** | **98.3** | **96.1** |
| LLaMA3.2-11B [Meta, 2024] | 82.3 | 66.1 | 70.6 | 69.1 | 75.9 | 66.2 | 71.1 | 67.6 | 85.8 | 89.5 | 90.7 | 88.9 |
| LLaMA3.2-90B [Meta, 2024] | 90.5 | 92.7 | 95.3 | 92.5 | 81.8 | 89.9 | 93.3 | 90.2 | 94.4 | 96.0 | 97.9 | 95.3 |
| Gemma3-1B [Google, 2025] | 46.7 | 47.4 | 54.7 | 53.9 | 46.7 | 46.7 | 55.7 | 53.7 | 67.5 | 66.9 | 68.5 | 65.6 |
| Gemma3-12B [Google, 2025] | 41.3 | 77.2 | 80.5 | 84.1 | 39.1 | 76.5 | 80.5 | 84.7 | 93.7 | 94.3 | 95.7 | 93.7 |
| Gemma3-27B [Google, 2025] | 44.0 | 80.4 | 81.9 | 85.7 | 45.6 | 80.7 | 80.1 | 85.0 | **95.7** | 96.1 | 96.9 | 95.9 |
| LLaVA1.6-7B [Liu et al., 2024b] | 69.3 | 53.4 | 57.7 | 54.6 | 68.7 | 47.3 | 54.7 | 48.7 | 76.1 | 76.2 | 79.2 | 74.4 |
| LLaVA1.6-13B [Liu et al., 2024b] | 67.7 | 76.7 | 81.3 | 79.9 | 64.5 | 72.5 | 81.2 | 75.7 | 85.1 | 87.5 | 90.3 | 85.9 |
| LLaVA1.6-34B [Liu et al., 2024b] | 82.7 | 84.4 | 88.3 | 86.0 | 73.4 | 83.4 | 89.3 | 86.3 | 92.5 | 93.0 | 94.5 | 91.9 |
| Pixtral-12B [Agrawal et al., 2024] | 87.5 | 90.0 | 90.5 | 90.1 | 72.1 | 88.3 | 91.4 | 89.9 | 92.9 | 94.4 | 95.6 | 93.1 |
| Phi4-5.6B [Microsoft, 2025] | 90.8 | 91.9 | 90.9 | 89.9 | 85.4 | 90.1 | 90.5 | 85.3 | 87.5 | 92.1 | 94.6 | 92.7 |
| BLIP3-4B [Xue et al., 2024] | 40.6 | 72.3 | 73.7 | 74.9 | 40.6 | 72.3 | 73.7 | 74.9 | 84.0 | 87.1 | 89.5 | 83.6 |

Table 5: Comparative evaluation of multiple vision-language models across real, synthetic, and textual modalities on four tasks. The best-performing result is highlighted in **bold**, and the second-best is underlined. Note that ER, RU, KG, and VR denote *entity recognition*, *relation understanding*, *knowledge grounding*, and *visual reasoning*, respectively.

 **C.3  Prompt Examples**

---

**Prompt for Diagram Cleaning**

**Positive Prompt:**

- A visual representation of information or data, explicitly intended for educational or scientific purposes. This includes flowcharts, circuit diagrams, architectural blueprints, and graphs, characterized by clear labeling and structured layout for easy understanding of complex concepts.

**Negative Prompts:**

- An **image of a company or brand logo**, designed to be a simple yet distinctive symbol that represents a company or product. Logos often consist of stylized letterforms, abstract geometric shapes, or a combination of both, and are designed to be easily recognizable even at small sizes. They usually feature a limited color palette and lack detailed textual information.

- An **image depicting natural landscapes**, including forests, mountains, rivers, or beaches, characterized by vivid natural colors and organic forms without any superimposed text or symbols.

- A **photograph** of one or several human beings, focusing on the face or figure, often capturing expression, personality, and mood, without any overlay of graphical information or text.

- **Images of old books, pages, or manuscripts**, primarily showing textual content in a historical or literary context, often with visible textures of paper and traditional fonts.

- A **screenshot** from a computer or mobile device, typically showing a user interface with icons, menus, and open applications, which may include web pages, software programs, or mobile apps.

- An **image with minimal visual content**, often appearing as a solid color background with sparse elements like one or two letters or one or two simple shapes. These images lack detail and complexity, presenting very basic or stark visual information with no significant features or recognizable patterns.

---

Figure 9: We perform six rounds of binary classification. In each round, an image is classified as a diagram or not by comparing its embedding with the embeddings of the two text prompts using MetaCLIP. Only images consistently classified as positive examples—that is, diagrams—across all rounds are retained.

## Prompt for Tagging (Step 1: Captioning)

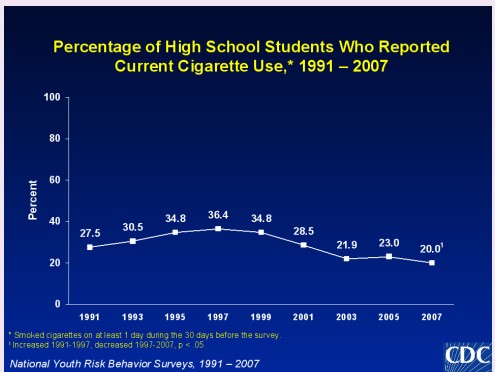

**System:** You are a diagram description assistant. Your task is to provide a detailed and structured description of the given diagram. Focus on aspects that might help to tag its domain (e.g., Biology, Chemistry, History) and type (e.g., Bar Chart, Flow Chart, Map).

**Context:** The diagram is sourced from Wikipedia, and here is some background information. Use the Wikipedia information above only if the diagram alone does not provide enough clarity or context. Always give priority to the information directly visible in the diagram for your analysis.

- **Page Title:** Prevalence of tobacco use.

- **Page Description:** Prevalence of tobacco use is reported by the World Health Organization, which focuses on cigarette smoking due to reported data limitations. Smoking has therefore been studied more extensively than any other form of consumption. Smoking is generally five times more prevalent among men than women; however, the gender gap differs across countries and is smaller in younger age groups. (text truncated due to space)

- **Diagram Description:** None.

**Instruction:** The description must be organized into the following three sections:

- **Content:** Describe key visual elements, labels, and any prominent features in the diagram.

- **Layout:** Explain how the elements are arranged (e.g., hierarchical, circular, linear) and the overall structure.

- **Function:** Indicate the likely purpose of the diagram (e.g., explaining a process, showing relationships, presenting data).

Figure 10: Before predicting tags for the diagrams, we conduct a captioning step. We instruct the VLM to act as a diagram description assistant and provide it with contextual information from Wikipedia, including the page title, page description, and diagram description (if available). The model is then prompted to focus on describing the content, layout, and function of the diagram.

## Prompt for Tagging (Step 2: Open-Ended Prediction)

**System:** You are a diagram tagging assistant. Your task is to analyze a diagram and identify its domain and type.

**Context:** The description of the diagram is provided for your reference:

- **Content:** The diagram appears to be a line graph depicting trends over time. It shows data points connected by lines, representing changes in a specific measure from 1991 to 2007. The graph includes numerical values on the y-axis and years on the x-axis. There are likely labels for the y-axis and x-axis, as well as a title at the top of the graph.

- **Layout:** The layout of the diagram is typical of a line graph. The vertical axis (y-axis) represents percentages, while the horizontal axis (x-axis) represents years. The data points are plotted along the x-axis and connected by lines to show the trend over time. The title is likely positioned at the top of the graph, providing context for the data being presented.

- **Function:** The function of this diagram is to visually represent and illustrate trends in a specific measure over a 16-year period. It allows viewers to quickly understand how the measured value has changed from 1991 to 2007. The use of a line graph makes it easy to see patterns, trends, and changes in the data over time, which is particularly useful for analyzing long-term data sets and identifying any significant shifts or fluctuations in the measured variable.

**Instruction:** Now analyze the diagram and provide its domain and type:

- **Domain:** The domain should be a specific field or area of knowledge. Its examples include Biology, Chemistry, Physics, Astronomy, History, etc.

- **Type:** The type should describe the nature of the diagram. Its examples include Bar Chart, Flow Chart, Table, Map, Logo, etc.

**Output Format:** Your output must be in the following JSON-like format. Do not provide any explanations or additional context. Only output the JSON object.

```
{
    "Domain": "string (must be 1 or 2 words)",
    "Type": "string (must be 1 or 2 words)"
}
```

Figure 11: After generating a caption for the diagram, we prompt the VLM again using the annotated content, layout, and function descriptions, and ask it to predict both a domain tag and a type tag. In this step, we adopt an open-ended setting, allowing the model to freely generate tags without any predefined options.

## Prompt for Tagging (Step 2: Multiple-Choice Prediction)

**System:** The same as Figure 11.

**Context:** The same as Figure 11.

**Instruction:** Now analyze the diagram and provide its domain and type:

- **Domain:** The domain should be a specific field or area of knowledge. Choose only one option from the following list:
  - Agriculture
  - Astronomy
  - Biology
  - Chemistry
  - Computer Science
  - Data Science
  - Environmental Science
  - Finance
  - Geography and Geology
  - Health Science
  - History
  - Mathematics
  - Music
  - Network Science
  - Operations Research
  - Physics
  - Political Science
  - Psychology
  - Sports
  - Transportation
  - Urban Planning

- **Type:** The type should describe the nature of the diagram. Choose only one option from the following list:
  - Bar Chart
  - Chemical Visual
  - Concept Diagram
  - Floor Plan
  - Flow Chart
  - Line Graph
  - Logo
  - Map
  - Network Chart
  - Pie Chart
  - Scatter Plot
  - Table
  - Technical Diagram
  - Timeline
  - Tree

**Output Format:** The same as Figure 11.

Figure 12: After generating open-ended tags, we apply clustering methods to analyze the tag distribution and identify a set of high-frequency tags, which are then used as options for the multiple-choice tagging setting. In this setting, we keep the instructions and context unchanged, but instead of allowing free predictions, the VLM is asked to select tags from the option list.

## Prompt for Statistical Annotation (Step 1: Captioning)

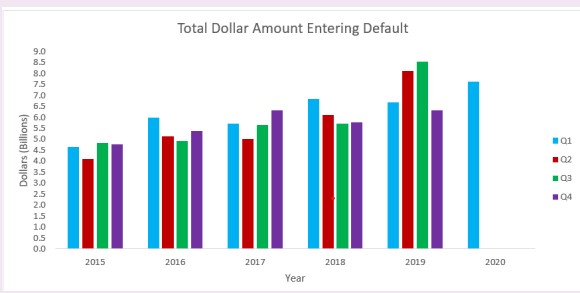

**System:** You are a scene graph construction assistant. Your task is to generate a detailed language-based description of a scene graph for a provided diagram.

**Context:** The diagram is sourced from Wikipedia, and here is some background information. Use the Wikipedia information above only if the diagram alone does not provide enough clarity or context. Always give priority to the information directly visible in the diagram for your analysis.

- **Page Title:** Federal Direct Student Loan Program.

- **Page Description:** The William D. Ford Federal Direct Loan Program provides low-interest loans for students and parents to help ... (text truncated due to space)

- **Diagram Description:** Total number of dollars (in billions) entering default, 2009-2018, data source: CRS.

**Instruction:**
- Identify key elements such as axes, labels, legends, colors, and numerical values.
- Describe trends, patterns, or outliers in the data, including peaks, or correlations.
- Explain relationships between different variables if applicable.
- Describe geographical features such as colored regions and arrows if applicable.
- Use clear and structured language.

**Examples:**
- The bar representing Q3 in 2019 is the tallest among all quarters.
- The blue line in the graph shows a steady increase from 2010 to 2018.
- The dark green segment in the pie chart represents 45.9 TWh of diesel consumption.
- The shaded region in the map highlights areas with the highest population density.
- The thick arrow marks the strongest southeastern wind current towards the country.

Figure 13: Similar to the tagging stage, we conduct a captioning step before generating semantic triples in order to reduce hallucinations. We also provide the model with contextual information from Wikipedia. For statistical diagrams, we instruct the model to focus on specific features such as numerical values and data trends. To enhance the quality of output, we manually design five descriptive sentences that serve as in-context examples during prompting.

## Prompt for Statistical Annotation (Step 2: Annotation)

**System:** You are an expert information extraction assistant specializing in scene graph construction. Your task is to analyze a given diagram description and extract meaningful, structured relationships between key elements.

**Context:** The description of the diagram is provided for your reference.

**1. Key Objects: X-axis:** Represents the years from 2009 to 2018. Each year is labeled along the axis. **Y-axis:** Represents the total dollars in billions entering default. The axis is labeled "Dollars in Billions". Numerical markers are present along the axis, though precise values are not clearly visible in the image. **Bars:** Vertical bars represent the amount of dollars entering default for each year. The height of each bar corresponds to the dollar amount. **Data Labels:** Numerical values are displayed above each bar, indicating the precise amount for each year.

**2. Attributes: X-axis:** Horizontal, evenly spaced tick marks representing years. **Y-Axes:** Vertical, with numerical markers indicating billions of dollars. The scale appears to range from approximately 0 to 80 billion. **Bars:** Vertical rectangular bars, colored blue. The width of each bar is uniform. **Data Labels:** Black text, positioned above each bar.

**3. Relationships:** Each bar is associated with a year on the x-axis and a value on the y-axis. The height of the bar corresponds directly to the value indicated by the data label and represents the amount in billions of dollars entering default in that year.

**4. Structural or Hierarchical Information:** The chart is a simple bar chart.

**5. Data Trends:** The chart shows a general trend of increasing dollars entering default from 2009 to a peak, followed by a decrease and then another increase toward the end of the period (2018). Precise yearly fluctuations are observable but require more detailed numerical data. There is no clear outlier year that significantly deviates from the general pattern.

**Instruction:**

- Identify important relationships between key elements from the description.
- Structure these relationships in the form of triples with three components:
    - **Source**: The primary element (subject) in the relationship.
    - **Relationship**: The type of connection between the source and target.
    - **Target**: The secondary element (object) in the relationship.
- Ensure that:
    - Each triple represents a meaningful connection between elements.
    - The relationships are concise yet descriptive.
    - There are no duplicate, redundant, or meaningless triples.

**Output Format:** The final output must strictly follow the JSON format below:
{
    "1": {"Source": "Triple 1", "Relationship": "Triple 1", "Target": "Triple 1"},
    ...
    "N": {"Source": "Triple N", "Relationship": "Triple N", "Target": "Triple N"}
}

Figure 14: After extracting relevant information from the diagram, we prompt the model to generate a list of triples, where each triple consists of a source (head entity), a relationship (relation), and a target (tail entity). To facilitate downstream processing, we instruct the model to produce the output in JSON format.

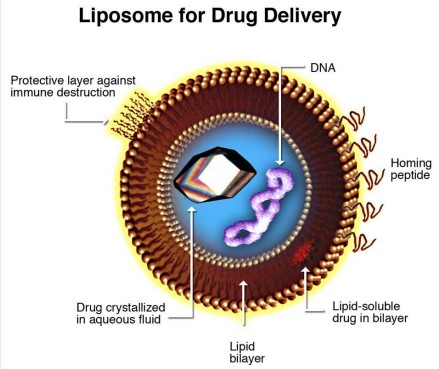

**System:** You are a scene graph construction assistant. Your task is to generate a detailed language-based description of a scene graph for a provided diagram.

**Context:** The diagram is sourced from Wikipedia, and here is some background information. Use the Wikipedia information above only if the diagram alone does not provide enough clarity or context. Always give priority to the information directly visible in the diagram for your analysis.

- **Page Title:** Nanomedicine.
- **Page Description:** Nanomedicine is the medical application of nanotechnology. Nanomedicine ranges from the medical applications of nanomaterials and biological devices, to nanoelectronic biosensors ... (text truncated due to space)
- **Diagram Description:** Liposomes are composite structures made of phospholipids and may contain small amounts of other molecules. Though liposomes can vary in size from low micrometer range to ... (text truncated due to space)

**Instruction:**
- Identify key objects, such as text, arrows, nodes, or data points.
- Identify attributes, such as size, color, shape, position, and numerical values.
- Explain how objects interact or relate to one another.
- Describe its overall hierarchy, structure or flow clearly if applicable.
- Use clear and structured language.

**Examples:**
- The newly discovered moon is connected to its elliptical orbit around Neptune.
- The blue alpha-helices are connected to beta-sheets through loop regions.
- The amine group ($-NH_2$) is added to the benzene ring at a new position.
- Each yellow triangular face is attached to three metallic rods at its edges.
- The E-flat note is positioned directly below the B-flat note on the staff.

Figure 15: The basic prompt framework for annotating scientific diagrams follows the same structure as that used for statistical diagrams. However, due to the inherent difference between scientific and statistical diagrams, we provide tailored instructions that emphasize features like objects, attributes, and structural hierarchy. We also include in-context examples specific to scientific content.

## Prompt for Scientific Annotation (Step 2: Annotation)

**System:** You are an expert information extraction assistant specializing in scene graph construction. Your task is to analyze a given diagram description and extract meaningful, structured relationships between key elements.

**Context:** The description of the diagram is provided for your reference.

The diagram depicts a liposome used for drug delivery. The central element is a large, circular liposome, predominantly brown-orange, representing a lipid bilayer. Inside the liposome, a light blue aqueous core contains a crystalline structure labeled "Drug crystallized in aqueous fluid" (white and iridescent) and a purple, coiled structure labeled "DNA". Several arrows connect labels to parts of the liposome:

- An arrow points from the text "Protective layer against immune destruction" to the outer edge of the liposome's lipid bilayer, indicating a protective function.
- Arrows point from the text "Lipid-soluble drug in bilayer" to the lipid bilayer itself, indicating the location of lipid-soluble drugs within the bilayer.
- Arrows point from the text "Drug crystallized in aqueous fluid" to the crystalline structure in the aqueous core.
- Arrows point from the text "Lipid bilayer" to the brown-orange lipid bilayer.

Attached to the outer edge of the liposome are several purple, wavy structures labeled "Homing peptide," suggesting a targeting mechanism. The text "Liposome for Drug Delivery" is positioned above the liposome, serving as a title. The overall structure is hierarchical, with the liposome as the central node, and various labels and arrows acting as connected nodes, describing its components and functions.

**Instruction:**

- Identify important relationships between key elements from the description.
- Structure these relationships in the form of triples with three components:
    - **Source**: The primary element (subject) in the relationship.
    - **Relationship**: The type of connection between the source and target.
    - **Target**: The secondary element (object) in the relationship.
- Ensure that:
    - Each triple represents a meaningful connection between elements.
    - The relationships are concise yet descriptive.
    - There are no duplicate, redundant, or meaningless triples.

**Output Format:** The final output must strictly follow the JSON format below:
{
    "1": {"Source": "Triple 1", "Relationship": "Triple 1", "Target": "Triple 1"},
    ...
    "N": {"Source": "Triple N", "Relationship": "Triple N", "Target": "Triple N"}
}

Figure 16: Similar to statistical diagrams, we provide the model with previously extracted information and ask it to generate a list of triples in JSON format.

## Prompt for QA Annotation (Step 1: Captioning)

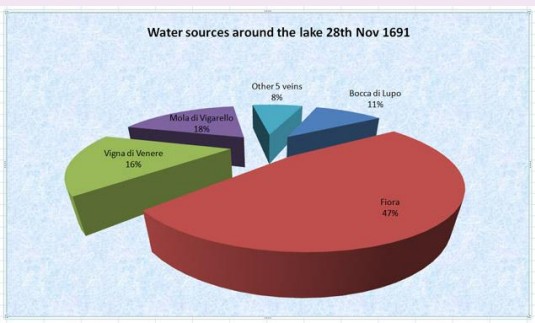

**System:** You are a diagram description assistant.

**Context:** The diagram is sourced from Wikipedia, and here is some background information. Use the Wikipedia information above only if the diagram alone does not provide enough clarity or context. Always give priority to the information directly visible in the diagram for your analysis.

- **Page Title:** Aqua Traiana.
- **Page Description:** The Aqua Traiana was a 1st-century Roman aqueduct built by Emperor Trajan and inaugurated on 24 June 109 AD. It channelled water from sources around Lake Bracciano, 40 kilometers north-west of Rome, to Rome in ancient Roman times but had fallen into disuse by the 17th century. (text truncated due to space)
- **Diagram Description:** None.

**Instruction:** Your task is to provide a detailed description of the diagram, addressing the following four aspects:

- **Recognition:** Identify and describe the key visual elements present in the diagram.
- **Understanding:** Explain the relationships and interactions between these elements.
- **Grounding:** Relate the diagram elements to real-world concepts or entities.
- **Reasoning:** Interpret the diagram to draw conclusions or infer information beyond what is explicitly shown.

**Output Format:** You must output your result in the following JSON-like format:
{
    "Recognition": "string or NA",
    "Understanding": "string or NA",
    "Grounding": "string or NA",
    "Reasoning": "string or NA"
}

Figure 17: Before annotating QA pairs, we prompt the model to caption the diagram. Here we provide relevant Wikipedia text and the definition of the four tasks to instruct the model to generate descriptions specific for QA annotation.

## Prompt for QA Annotation (Step 2: Annotation)

**System:** You are a question-answering annotation assistant. Your task is to analyze a diagram and annotate question-answering pairs.

**Context:** The description of the diagram is provided for your reference.

**1. Recognition:** The diagram is a 3D pie chart showing the distribution of water sources around a lake on November 28th, 1691. The chart is segmented into six sections, each representing a different water source and its percentage contribution. The sections are color-coded for easy identification. The labels for each section indicate the name of the water source and its corresponding percentage. The largest section is labeled "Fiora" and occupies 47% of the chart. The other sections are "Vigna di Venere" (16%), "Mola di Vigarello" (18%), "Bocca di Lupo" (11%), "Other 5 veins" (8%).

**2. Understanding:** The pie chart illustrates the relative proportions of water sourced from different locations around the lake on the specified date. Each slice represents a specific water source, and its size is proportional to its contribution to the total water supply. The chart visually compares the contributions of each source, highlighting the dominance of Fiora with 47% of the total water supply.

**3. Grounding:** The diagram represents the real-world distribution of water sources around a lake (likely Lake Bracciano based on the Wikipedia context) at a specific point in time. The named sources ("Fiora", "Vigna di Venere", etc.) are likely geographical locations or specific water channels feeding into the lake. The percentages represent the proportion of the total water volume coming from each source.

**4. Reasoning:** The chart shows that Fiora is the most significant water source, contributing almost half (47%) of the total water. The remaining sources contribute smaller, but still substantial, portions. This suggests that the water supply relied heavily on Fiora, while other sources played supporting roles. The data could be used to understand the hydrological characteristics of the area and the historical reliance on different water sources. The date (28th Nov 1691) implies this is a snapshot of a specific time and conditions, and the distribution might vary across seasons or years.

Figure 18: After captioning, we provide the model with the extracted information corresponding to the four tasks.

## Prompt for QA Annotation (Step 2: Annotation) (Continued)

**Instruction:** Create four multiple-choice questions based on the diagram, each focusing on one of the four aspects:

- **Recognition**: Test the identification of elements in the diagram.
- **Understanding**: Assess comprehension of the relationships or processes depicted.
- **Grounding**: Evaluate the ability to connect elements to real-world knowledge.
- **Reasoning**: Challenge inference or prediction based on the diagram.

For each question:

- Provide a clear question statement.
- Offer exactly four options labeled A, B, C, and D.
- Indicate the correct answer, which must be only one among A, B, C, or D.

**Output Format:** You must output your result in the following JSON-like format:

```
{
    "Recognition": {
        "Question": "string",
        "Options": { "A": "string", "B": "string", "C": "string", "D": "string" },
        "Answer": "A/B/C/D"
    },
    "Understanding": {
        "Question": "string",
        "Options": { "A": "string", "B": "string", "C": "string", "D": "string" },
        "Answer": "A/B/C/D"
    },
    "Grounding": {
        "Question": "string",
        "Options": { "A": "string", "B": "string", "C": "string", "D": "string" },
        "Answer": "A/B/C/D"
    },
    "Reasoning": {
        "Question": "string",
        "Options": { "A": "string", "B": "string", "C": "string", "D": "string" },
        "Answer": "A/B/C/D"
    }
}
```

Figure 19: Using the descriptive information, we instruct the model to generate one multiple-choice question for each of the four tasks. Each question is designed to include exactly four answer options with a single correct answer.

## QA Annotation Example

**Recognition**: What percentage of water sources around the lake on November 28th, 1691, came from Bocca di Lupo?
A) 11%
B) 8%
C) 16%
D) 18%
**Answer:** A

**Understanding**: What does the size of each segment in the pie chart represent?
A) The geographical location of the water source.
B) The depth of the water source.
C) The volume of water contributed by each source relative to the total.
D) The water temperature of each source.
**Answer:** C

**Grounding**: Considering the context of the chart, which of the following is MOST likely a real-world interpretation of Fiora, Vigna di Venere, and Mola di Vigarello?
A) Types of water purification methods used in 1691.
B) Names of individuals who managed the water supply.
C) Geographical locations or water channels supplying the lake.
D) Measurements of water flow rates.
**Answer:** C

**Reasoning**: Based on the data, which of the following conclusions is MOST supported?
A) The water supply was evenly distributed among all sources.
B) Fiora was the primary source of water for the lake.
C) The "Other 5 veins" contributed the least amount of water, rendering them insignificant.
D) Vigna di Venere was the most important water source besides Fiora.
**Answer:** B

Figure 20: Here we illustrate an example of the annotated results, including questions, options, and correct answers.

## Pipeline for Benchmark Evaluation

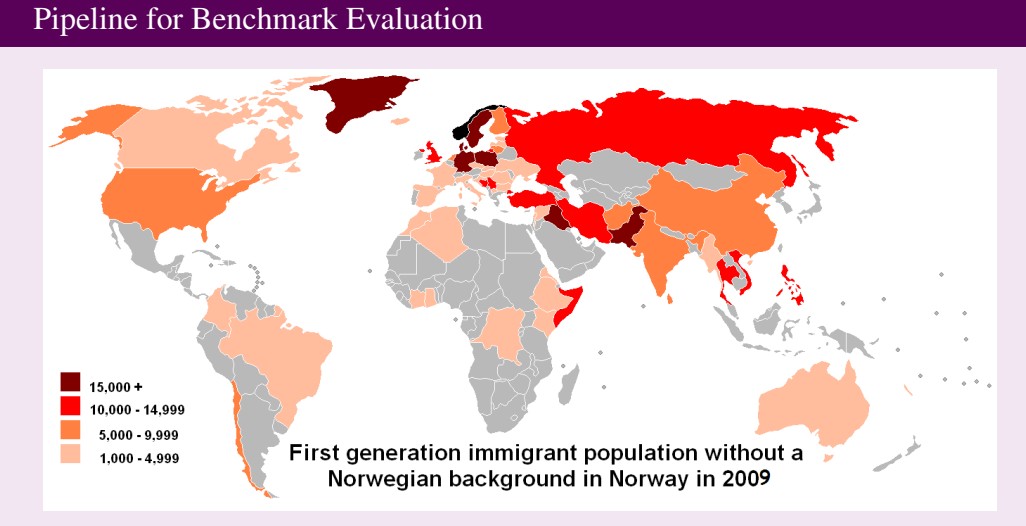

**Step 1: Analysis**
**Prompt:**
You are a Visual Question Answering expert. Based on the image, analyze and answer the following question:
What color represents countries with 10,000-14,999 first-generation immigrants in 2009?

A) Dark Red.
B) Red.
C) Light Orange.
D) Light Peach.

**Output:**
Okay, I've analyzed the image. Based on the legend provided, countries with 10,000-14,999 first-generation immigrants in Norway in 2009 are represented by **B. Red**. Let me know if you have another question about the image!

**Step 2: Conclusion**
**Prompt:**
Based on the context above, give your final choice. Do not provide any explanations.
**Output:**
The final choice is B.

**Step 3: Choice**
**Input:** The final choice is B.
**Output:** B

Figure 21: This figure illustrates the pipeline used for benchmark evaluation. First, the model is prompted to analyze the input data and the question. Next, it is asked to produce a final conclusion based on its analysis. Finally, a response-processing step is applied to extract the answer from the conclusion text.

