# OpenReview forum: "Chimera: Revisiting Shortcut Learning by Vision-Language Models for Diagram Understanding"
_NeurIPS.cc/2025/Datasets_and_Benchmarks_Track — Submitted to NeurIPS 2025 Datasets and Benchmarks Track_

### Official Review · Reviewer_8MPb · 2025-06-08

**Rating:** 5
**Confidence:** 4

**Summary:**

The paper introduces CHIMERA, a benchmark collecting 7,500 diagrams with semantic triple annotations and multi-level questions, designed to evaluate genuine diagram comprehension in VLMs by assessing core skills.

It identifies three key shortcuts inflating VLM performance: (1) visual-memorization (slight impact), (2) knowledge-recall (moderate role), and (3) Clever-Hans (major reliance on linguistic priors), with experiments showing VLMs often exploit superficial and explicit cues rather than true multimodal reasoning.

The study reveals that current VLMs’ strong benchmark performance masks fundamental flaws in diagram understanding, urging the field to prioritize robust evaluation frameworks over inflated metrics.

**Dataset Code Accessibility:**

Yes

**Ethical Considerations:**

No, there are no or only very minor ethics concerns

**Limitations Weaknesses:**

CHIMERA relies heavily on Wikipedia diagrams, which may limit its generalizability to extend such a method to other use cases. For example, by collecting diagrams from a lot more resources e.g. lucid chart, google diagrams, e.t.c. this work would be more meaningful.

The paper also relies heavily on Gemini for generating semantic triple annotations, which may inherit biases or inaccuracies from the model itself. For example, using other models such as GPT, Qwen e.t.c to produce a cross-table comparison between the results may show off the true cross-reasoning ability of VLMs with-in or with-out themselves.

**Strengths Contributions:**

The design of the benchmark is innovative: by annotating diagrams with semantic triples, CHIMERA bridges visual, symbolic, and textual modalities, enabling rigorous analysis of how models derive answers—not just whether they are correct.

With 7,500 diverse, high-quality diagrams and controlled annotation pipelines, CHIMERA sets a new standard for dataset rigor.

The Clever-Hans experiments (e.g., removing diagrams) are particularly compelling, exposing how heavily models rely on linguistic biases rather than visual reasoning.

---

> ### Author Rebuttal · Authors · 2025-07-28
>
> We sincerely thank the reviewer for recognizing the novelty of CHIMERA’s design, the rigor of our annotation pipeline, and the value of our shortcut diagnostic experiments. We greatly appreciate the thoughtful feedback, and we address the reviewer’s concerns in detail below.
>
> >**Point 1**: CHIMERA relies heavily on Wikipedia diagrams, which may limit its generalizability to extend such a method to other use cases. For example, by collecting diagrams from a lot more resources e.g. lucid chart, google diagrams, e.t.c. this work would be more meaningful.
>
> **Response**: We thank the reviewer for raising this important point. We agree that expanding beyond Wikipedia to include diagrams from platforms like Lucidchart or Google Drawings could further enhance domain coverage and practical relevance. Our current choice of Wikipedia was motivated by its open-access nature, broad topical diversity, and ease of large-scale crawling, which makes it an ideal foundation for reproducible benchmarking. That said, a key contribution of CHIMERA lies in its modular, domain-agnostic pipeline, which can be applied to other diagram sources with minimal modification. We view our work as a first step toward broad-coverage diagram understanding, and we hope it will inspire future extensions into more specialized and professionally-oriented domains.
>
>
> >**Point 2**: The paper also relies heavily on Gemini for generating semantic triple annotations, which may inherit biases or inaccuracies from the model itself. For example, using other models such as GPT, Qwen e.t.c to produce a cross-table comparison between the results may show off the true cross-reasoning ability of VLMs with-in or with-out themselves.
>
> **Response**: We appreciate the reviewer’s insightful suggestion. We fully agree that relying on a single model for annotation may introduce model-specific biases. However, several considerations informed our decision to use Gemini-2.0-Flash:
> - It had strong performance at the time of annotation, especially for multimodal tasks.
> - It operated under an annotation-only setting, with access to the full Wikipedia page context and diagram description, which is withheld during model evaluation, thus avoiding leakage or unfair advantage.
> - Its outputs were post-processed through rigorous multi-stage filtering and human validation to reduce potential hallucinations or artifacts.
>
> We also recognize the value of cross-model comparison for semantic annotation. However, due to the large dataset scale (7.5k diagrams), this would require considerable computational resources. As noted in Appendix C.1.4 Table 4, annotating with just one model already costs over $20,000 USD, making multi-model annotation currently infeasible for our setting. That said, our annotation pipeline is model-agnostic and easily supports future re-annotation or cross-validation with alternative models such as GPT or Qwen. We believe CHIMERA can serve as a reusable infrastructure for such future efforts.

---

### Official Review · Reviewer_JmgV · 2025-07-02

**Rating:** 5
**Confidence:** 4

**Summary:**

This paper introduces Chimera, a benchmark for diagnosing shortcut learning in vision-language models on diagram understanding tasks. Using minimal pairs, it reveals models' overreliance on visual or structural cues over semantic understanding and proposes prompting strategies to mitigate such behavior.

**Dataset Code Accessibility:**

Yes

**Ethical Considerations:**

No, there are no or only very minor ethics concerns

**Final Justification:**

The authors have addressed my concern and promised to include related works in their revised version of the main paper.

**Limitations Weaknesses:**

While the supplementary material includes a list of related work, the main paper lacks a dedicated Related Work section. This makes it difficult for readers to contextualize the Chimera benchmark within prior research on shortcut behaviors and LLM evaluation. Including a concise but informative discussion of related benchmarks and prior shortcut analyses in the main text would significantly enhance clarity and strengthen the paper’s positioning in the literature.

**Strengths Contributions:**

- The paper is, to the best of my knowledge, the first to systematically investigate shortcut learning in diagram-based reasoning tasks.
- While shortcut behaviors have been widely studied in language-only settings (e.g., NLI, QA), this work expands the scope to multimodal reasoning.
- The benchmark design isolates subtle differences in diagram inputs to reveal when models overfit to non-semantic cues. The analysis shows that even strong models (e.g., GPT-4V, Claude) can fail to generalize in structured visual contexts.

---

> ### Author Rebuttal · Authors · 2025-07-28
>
> We thank the reviewer for recognizing the novelty of our work in systematically investigating shortcut behaviors in multimodal settings, especially diagram-based reasoning. We also appreciate the acknowledgment of our benchmark design and its ability to reveal non-semantic biases even in strong models. Below, we address the reviewer’s concern about the discussion of related work.
>
> >**Point 1**: The main paper lacks a dedicated Related Work section. This makes it difficult for readers to contextualize the Chimera benchmark within prior research on shortcut behaviors and LLM evaluation. Including a concise but informative discussion of related benchmarks and prior shortcut analyses in the main text would significantly enhance clarity and strengthen the paper’s positioning in the literature.
>
> **Response**: We thank the reviewer for pointing out this important issue. We agree that a clear comparison with prior benchmarks and shortcut studies in the main paper would enhance clarity and strengthen our contributions.
> To address this, we will revise the camera-ready version to explicitly include a concise Related Work section. This section will compare CHIMERA with existing diagram understanding benchmarks (e.g., ChartQA, AI2D, TQA), highlighting key differences in scale, domain diversity, annotation granularity, shortcut mitigation, and multimodal evaluation.
> CHIMERA is, to our knowledge, the first benchmark to:
>
> - Introduce multi-level QA tasks reflecting distinct stages of diagram comprehension;
> - Include shortcut-aware annotation filtering to avoid language-only biases;
> - Provide structured semantic annotations (triples) and support three modality inputs (visual, semantic, textual).
>
> We provide a summary comparison table below to better illustrate CHIMERA’s novelty:
>
> | Benchmark   | Size  | Domain Coverage | Multi-level QA | Shortcut Filtering | Diagram Info Annotation | Multimodal Inputs             |
> | ----------- | ----- | ------------------ | -------------- | ------------------ | ----------------------- | ----------------------------- |
> | ChartQA     | \~20K | Charts only        | ✗              | ✗                  | ✗                       | ✗                             |
> | FigureQA    | 100K+ | Synthetic figures  | ✗              | ✗                  | ✗                       | ✗                             |
> | FlowChartQA | \~5K  | Flowcharts         | ✗              | ✗                  | ✗                       | ✗                             |
> | AI2D        | \~5K  | Elementary science | ✗              | ✗                  | Partial (scene graph)   | ✗                             |
> | ScienceQA   | \~21K | Science diagrams   | ✗              | ✗                  | ✗                       | ✗                             |
> | TQA         | \~16K | Textbook diagrams  | ✗              | ✗                  | ✗                       | ✗                             |
> | **CHIMERA** | 7.5K  | Broad (Wikipedia)  | ✓              | ✓                  | ✓ (semantic triples)    | ✓ (visual, semantic, textual) |
>
> We hope this detailed comparison clarifies how CHIMERA advances the state of diagram understanding benchmarks and serves as a foundation for future multimodal evaluation and diagnostic efforts.

---

### Official Review · Reviewer_dRSG · 2025-07-03

**Rating:** 5
**Confidence:** 3

**Summary:**

This paper introduces CHIMERA, a new benchmark designed to rigorously evaluate the diagram understanding abilities of Vision-Language Models (VLMs). The benchmark consists of 7,500 high-quality diagrams from Wikipedia, each annotated with semantic triples and multi-level questions that assess four aspects: entity recognition, relation understanding, knowledge grounding, and visual reasoning. The authors use CHIMERA to systematically investigate three types of shortcut behaviors (visual-memorization, knowledge-recall, and Clever-Hans) that may inflate model performance without genuine comprehension. Through comprehensive experiments on 15 open-source VLMs from 7 model families, the study reveals that current VLMs’ apparent strong performance on diagram-based VQA tasks is largely driven by these shortcuts—especially Clever-Hans and knowledge-recall—rather than true multimodal reasoning.

**Dataset Code Accessibility:**

Yes

**Ethical Considerations:**

No, there are no or only very minor ethics concerns

**Limitations Weaknesses:**

- While the authors apply multi-stage consistency checks and human evaluation, much of the annotation—especially for complex reasoning questions—relies on automated models (e.g., Gemini-2.0-Flash). Subtle errors, model biases, or annotation artifacts may persist, particularly for questions that require nuanced interpretation, which could impact evaluation reliability
- The shortcut analysis, although systematic, is mostly correlational: it demonstrates the presence of shortcut behaviors but does not provide causal or mechanistic explanations.

**Strengths Contributions:**

- The authors construct CHIMERA with a strong semiotic foundation, providing detailed annotations in three modalities (visual, semantic, textual) and four tasks that map onto key stages of diagram comprehension. The benchmark covers diverse diagram types and includes rigorous multi-stage quality checks, including human evaluation, to ensure annotation quality
- The paper makes a significant methodological contribution by precisely defining and empirically evaluating three types of shortcut behaviors (visual-memorization, knowledge-recall, and Clever-Hans). The experimental protocol directly tests for these shortcuts using ablation (e.g., removing visual input), which gives clear diagnostic insight into what current VLMs are (not) learning
- This paper is well written and easy to follow.
- The authors provide very detail inter-annotation statisitic for the annotation quality assessment.

---

> ### Author Rebuttal · Authors · 2025-07-28
>
> We would like to thank the reviewer for the thoughtful and encouraging feedback. We appreciate the recognition of our benchmark's strong semiotic foundation, the multimodal and multi-stage design of CHIMERA, and the rigorous quality control of our annotations. We are also glad that the reviewer found our analysis of shortcut behaviors to be a meaningful methodological contribution, and appreciated the clarity of our writing and the detailed inter-annotator statistics. Below, we respond to the two main concerns.
>
> >**Point 1**: *While the authors apply multi-stage consistency checks and human evaluation, much of the annotation—especially for complex reasoning questions—relies on automated models (e.g., Gemini-2.0-Flash). Subtle errors, model biases, or annotation artifacts may persist, particularly for questions that require nuanced interpretation, which could impact evaluation reliability.*
>
> **Response**: We appreciate the reviewer’s thoughtful concern about annotation reliability. While we use Gemini-2.0-Flash to assist with initial annotation, it operates under a different setting from model evaluation: it has access to rich background context, including Wikipedia article text and diagram descriptions, which are hidden during testing. This significantly reduces risk of bias or shortcut during annotation.
> In addition, we apply multi-stage quality control, including consistency checks across modalities and manual verification, especially for reasoning questions, to filter out ambiguous or biased cases. While some residual noise is inevitable, we believe this pipeline offers a strong balance between quality and scalability. Looking forward, our annotation framework can easily incorporate stronger models (e.g., Gemini-2.5-Pro) for even more reliable benchmarks.
>
>
> >**Point 2**: *The shortcut analysis, although systematic, is mostly correlational: it demonstrates the presence of shortcut behaviors but does not provide causal or mechanistic explanations.*
>
> **Response**: We appreciate the reviewer’s thoughtful observation. As also noted in our response to Reviewer RfXX, our work is diagnostic in nature: it aims to reveal the prevalence and forms of shortcut behaviors through controlled evaluation, rather than to pinpoint their internal causes. We agree that understanding the causal origins of these behaviors, such as training data biases, architectural constraints, or modality imbalances, is an important and timely research direction. However, such investigation requires detailed probing of model internals, which is beyond the scope of this paper. We hope CHIMERA’s structured design, particularly its minimal-input variants and modality-controlled tasks, offers a strong foundation for future research on causal and mechanistic explanations.

---

### Official Review · Reviewer_RfXX · 2025-07-05

**Rating:** 5
**Confidence:** 4

**Summary:**

This paper introduces CHIMERA, a comprehensive benchmark designed to evaluate the performance of vision-language models (VLMs) in diagram understanding. CHIMERA comprises 7,500 high-quality diagrams, each annotated with semantic triples and multi-level questions targeting four key aspects: entity recognition, relation understanding, knowledge grounding, and visual reasoning. Through experiments, the paper identifies three types of shortcut behaviors in VLMs—visual-memorization, knowledge-recall, and Clever-Hans shortcuts—and demonstrates that current models rely heavily on these shortcuts rather than genuine comprehension.

**Dataset Code Accessibility:**

Yes

**Dataset Code Comments:**

The dataset is accessable and reproducible.

**Ethical Considerations:**

No, there are no or only very minor ethics concerns

**Final Justification:**

After reading the authors' responses and other reviewers' comments, I will keep my original rating.

**Limitations Weaknesses:**

1. CHIMERA primarily sources diagrams from Wikipedia, which may not fully represent specialized domains (e.g., medicine, engineering), limiting the benchmark's generalizability.
2. While the paper identifies three shortcut behaviors, it lacks an in-depth exploration of their underlying causes (e.g., model architecture or training data biases).
3. This paper does not provide a detailed comparison between CHIMERA and current diagram understanding benchmarks, such as data volume, annotation methods, or dataset characteristics (e.g., domain coverage, task granularity). This omission makes it difficult to assess CHIMERA’s novelty and advantages over prior work.
4. This paper focuses on exposing problems but does not propose concrete solutions or model optimization strategies to address the identified shortcuts. It is recommended to outline the general strategy or indicate that specific methodological research will be conducted in the future work to address the issues identified in this paper.

**Strengths Contributions:**

1. This paper is well-organized and easy to understand.
2. CHIMERA is grounded in semiotic theory, offering a multi-modal (visual, semantic, textual) and multi-stage (four comprehension tasks) evaluation framework, which provides a thorough analysis of diagram understanding.
3. This paper ensures high-quality annotations through human evaluation and consistency checks.
4. This paper presents convincing experimental results for CHIMERA by evaluating 15 open-source VLMs.
5. This study highlights critical limitations of current VLMs and offers valuable insights for future model development, particularly in mitigating shortcut behaviors and enhancing true comprehension.

---

> ### Author Rebuttal · Authors · 2025-07-28
>
> We sincerely thank the reviewers for recognizing the semiotic foundation of our benchmark design, as well as the quality of our annotations and experimental results. Below, we respond to each concern in detail.
>
> >**Point 1**: *CHIMERA primarily sources diagrams from Wikipedia, which may not fully represent specialized domains (e.g., medicine, engineering), limiting the benchmark's generalizability.*
>
> **Response**: We appreciate this thoughtful observation. While we acknowledge that Wikipedia may not fully capture highly specialized domains, it remains one of the most diverse and publicly accessible sources of diagrammatic knowledge, covering a broad range of topics. Our goal was to ensure coverage across general-purpose domains while maintaining transparency and reproducibility. Importantly, our data curation and annotation pipeline, described in detail in the paper, is modular and domain-agnostic, making it readily extensible to specialized or sensitive domains in future work.
>
>
>
> >**Point 2**: *While the paper identifies three shortcut behaviors, it lacks an in-depth exploration of their underlying causes (e.g., model architecture or training data biases).*
>
> **Response**: We thank the reviewer for this insightful and fundamental point. Our current work focuses on systematically exposing the limitations of VLMs through shortcut behaviors, revealing that their success often stems from superficial patterns rather than genuine diagram understanding. We agree that uncovering the underlying causes, such as biases in training data, limitations in model architecture, or insufficient cross-modal supervision, is a crucial next step. However, this is a broad and complex question that goes beyond the scope of this paper. We hope our benchmark provides a diagnostic tool to inspire future research into mechanistic interpretations and improvements of VLMs.
>
>
> >**Point 3**: *This paper does not provide a detailed comparison between CHIMERA and current diagram understanding benchmarks, such as data volume, annotation methods, or dataset characteristics (e.g., domain coverage, task granularity). This omission makes it difficult to assess CHIMERA’s novelty and advantages over prior work.*
>
> **Response**: We thank the reviewer for this important observation. We agree that a clearer positioning of CHIMERA with respect to prior benchmarks would help better highlight its novelty. To this end, we clarify as follows that our benchmark differs from existing diagram understanding benchmarks along several dimensions:
>
> - Scale and Domain Coverage: CHIMERA contains 7,500 diagrams filtered from millions of images on Wikipedia, covering a broad range of domains beyond those in existing benchmarks.
> - Multi-level Comprehension: CHIMERA introduces a structured four-level QA design, entity recognition, relation understanding, knowledge grounding, and visual reasoning, which reflects a more comprehensive view of diagram understanding.
> - Shortcut-Free QA Design: To mitigate shortcut artifacts, we apply multi-stage filtering (including diagram ablations and answer consistency checks) to reduce the likelihood of language-only or template-based shortcut exploitation.
> - Diagram Information Annotation: We annotate each diagram with a set of semantic triples, enabling interpretable, symbolic representations of visual content.
> - Multimodal Inputs: CHIMERA supports evaluation across three modalities, visual (raw diagrams), semantic (visualized triples), and textual (sentence-form triples), enabling fine-grained analysis of modality alignment.
>
> We provide a qualitative comparison of existing benchmarks below:
>
> | Benchmark   | Size  | Domain Coverage  | Multi-level QA | Shortcut Filtering | Diagram Info Annotation | Multimodal Inputs             |
> | ----------- | ----- | ------------------- | -------------- | ------------------ | ----------------------- | ----------------------------- |
> | ChartQA     | \~20K | Charts only         | ✗              | ✗                  | ✗                       | ✗                             |
> | FigureQA    | 100K+ | Synthetic figures   | ✗              | ✗                  | ✗                       | ✗                             |
> | FlowChartQA | \~5K  | Flowcharts          | ✗              | ✗                  | ✗                       | ✗                             |
> | AI2D        | \~5K  | Elementary science  | ✗              | ✗                  | Partial (scene graph)   | ✗                             |
> | ScienceQA   | \~21K | Science diagrams    | ✗              | ✗                  | ✗                       | ✗                             |
> | TQA         | \~16K | Textbooks (science) | ✗              | ✗                  | ✗                       | ✗                             |
> | **CHIMERA** | 7.5K  | Broad (Wikipedia)   | ✓              | ✓                  | ✓ (semantic triples)    | ✓ (visual, semantic, textual) |
>
> We will revise the Related Work section in the camera-ready version to explicitly include this comparison.
>
>
> >**Point 4**: *This paper focuses on exposing problems but does not propose concrete solutions or model optimization strategies to address the identified shortcuts. It is recommended to outline the general strategy or indicate that specific methodological research will be conducted in the future work to address the issues identified in this paper.*
>
> **Response**: We appreciate the reviewer’s suggestion. Our work aims to lay the foundation for understanding the limitations of current VLMs in diagram comprehension by rigorously diagnosing shortcut behaviors, an essential first step before designing effective solutions. Given the complexity and generality of shortcut learning across models, modalities, and tasks, we believe that providing a universally effective solution requires broad community effort and multiple research directions.
> That said, our benchmark offers not just evaluation signals but also structured supervision (e.g., semantic triples, multi-modal inputs) that can be leveraged for future model development. We hope CHIMERA can serve as both a diagnostic tool and a resource to guide future work in improving model robustness, multimodal alignment, and symbolic reasoning. We see this paper as a call to action and look forward to exploring these questions further in follow-up work.

---

### Decision · Program_Chairs · 2025-09-18

**Decision:**

Reject

**Comment:**

The paper presents a novel benchmark CHIMERA focused on diagram understanding. While all reviewers recommend acceptance, the 2 primary issues raised are a lack of diversity in the images as the benchmark is sourced entirely from Wikipedia, and uses synthetic responses from Gemini for annotations. While I am leaning in favor of accepting this paper, I wholeheartedly agree with these 2 valid criticisms raised, and would like the authors to concretely address the two issues prior to publication. Specifically, they should rename / modify the title of the benchmark and description to *clearly indicate* that this benchmark is measuring diagram understanding on Wikipedia images (e.g., call it WikiDiagrams for example), and secondly, manually inspect a subset of responses to identify the error rate in the ground truth.

Most frontier LLMs are trained with Wikipedia data and hence this benchmark is almost certainly contaminated with training sets of frontier models. Having these changes made up front and visible would ensure that this does not become a train set accuracy measurement.

I recommend poster acceptance but only once these necessary changes are made.

===== FINAL UPDATE FROM DB Track PCs ====

The final decision for this paper has been taken by the program chairs after consultation with the SACs. All Senior Area Chairs have ranked papers according to the feedback from the AC during the review process. We decided to leave the original meta-review to reflect the opinion of the AC in light of the initial discussions with reviewers and SAC.